# Neonatal azithromycin administration to prevent infant mortality: study protocol for a randomised controlled trial

Ali Sie,[1] Mamadou Bountogo,[1] Eric Nebie,[1] Mamadou Ouattara,[1] Boubacar Coulibaly,[1] Cheik Bagagnan,[1] Pascal Zabre,[1] Elodie Lebas,[2] Jessica Brogdon,[2] William W Godwin,[2] Ying Lin,[2] Travis Porco,[2,3,4] Thuy Doan,[2,3] Thomas M Lietman,[2,3,4] Catherine E Oldenburg,[2,3,4] NAITRE Study Group

¹Centre de Recherche en Sante de Nouna, Nouna, Burkina Faso
²Francis I Proctor Foundation, University of California, San Francisco, California, USA
³Department of Ophthalmology, University of California, San Francisco, California, USA
⁴Department of Epidemiology and Biostatistics, University of California, San Francisco, California, United States

**Correspondence to**
Dr Catherine E Oldenburg;
catherine.oldenburg@ucsf.edu

## ABSTRACT

**Introduction** Biannual mass azithromycin distribution to children aged 1–59 months has been shown to reduce all-cause mortality. Children under 28 days of age were not treated in studies evaluating mass azithromycin distribution for child mortality due to concerns related to infantile hypertrophic pyloric stenosis (IHPS). Here, we report the design of a randomised controlled trial to evaluate the efficacy and safety of administration of a single dose of oral azithromycin during the neonatal period.

**Methods and analysis** The *Nouveaux-nés et Azithromycine: une Innovation dans le Traitement des Enfants* (NAITRE) study is a double-masked randomised placebo-controlled trial designed to evaluate the efficacy of a single dose of azithromycin (20 mg/kg) for the prevention of child mortality. Newborns (n=21 712) aged 8–27 days weighing at least 2500 g are 1:1 randomised to a single, directly observed, oral dose of azithromycin or matching placebo. Participants are followed weekly for 3 weeks after treatment to screen for adverse events, including IHPS. The primary outcome is all-cause mortality at the 6-month study visit.

**Ethics and dissemination** This study was approved by the Institutional Review Boards at the University of California, San Francisco in San Francisco, USA (Protocol #18-25027) and the Comité National d'Ethique pour la Recherche in Ouagadougou, Burkina Faso (Protocol #2018-10-123). The findings of this trial will be presented at local, regional and international meetings and published in open access peer-reviewed journals.

**Trial registration number** NCT03682653; Pre-results.

## INTRODUCTION

The MORDOR study demonstrated that mass azithromycin distribution reduces all-cause child mortality relative to placebo in high mortality settings in sub-Saharan Africa.[1 2] The largest effects were seen in the youngest children, with nearly 25% reduction in mortality in children aged 1–5 months. Children in this age range have the highest risk of mortality

## Strengths and limitations of this study

► This study is a large individually randomised placebo-controlled trial evaluating a single dose of azithromycin compared with a single dose of matching placebo during the neonatal period for the prevention of infant mortality.

► This study will use an identical matching placebo, which will minimise bias between the two arms, and participants, outcome assessors and investigators are masked to treatment allocation.

► Infants will be carefully monitored for adverse events following treatment administration to screen for infantile hypertrophic pyloric stenosis (IHPS) and other potential adverse events.

► Limitations of this study include the limited number of assessments that can be made with a large sample size and that the study may be underpowered to detect IHPS, given the expected rarity of the condition.

and thus may have the most to gain from child survival interventions. MORDOR treated children as young as 1 month of age due to increased mortality in this population but did not treat children before 28 days of life due to concerns related to potentially increased risk of infantile hypertrophic pyloric stenosis (IHPS) related to macrolide use.[3 4] No cases of IHPS or increases in vomiting were found in infants aged 1–5 months receiving azithromycin in MORDOR relative to placebo.[5]

Globally, reductions in neonatal mortality have been slower than reduction in post-neonatal mortality, and slow progress in reduction in neonatal mortality has affected overall progress in achieving under-five mortality targets.[6 7] The majority of neonatal deaths occur during the first week of life, a majority of which are due to preterm birth and intrapartum-related conditions.[7 8] Neonatal mortality is more

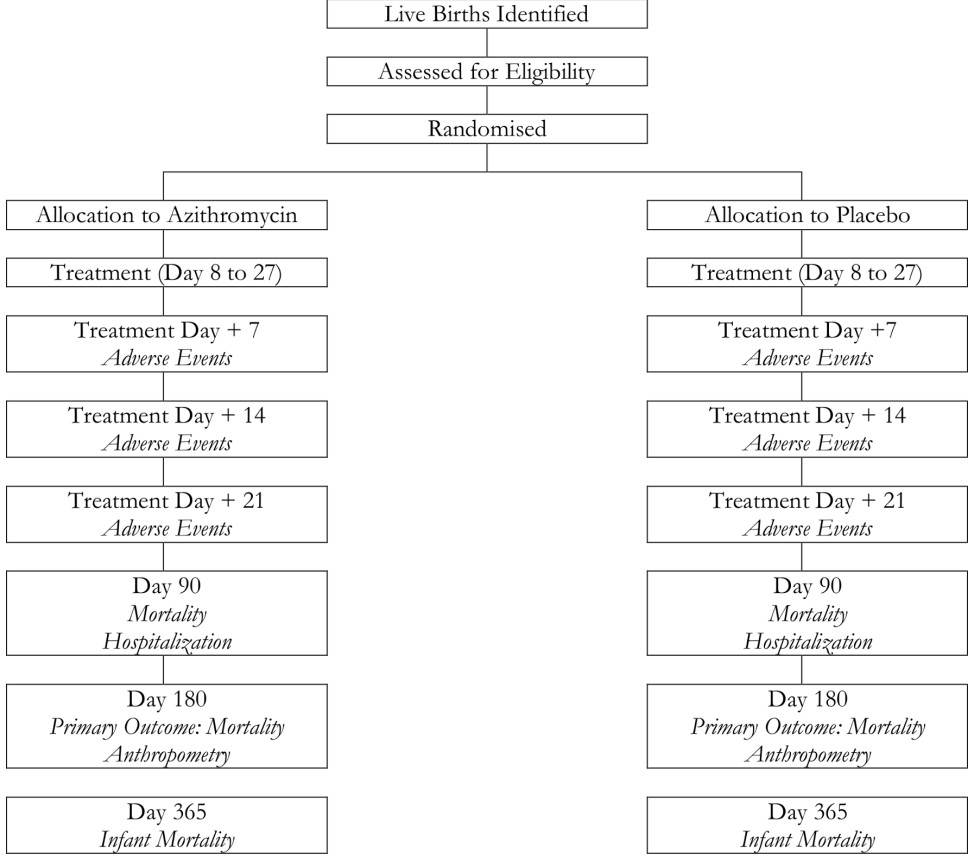

**Figure 1** Study flow diagram for study visits and outcome assessments.

likely to be infectious in nature in the late neonatal period (8–27 days), during which time the most common cause of neonatal mortality is sepsis.[7] Post-neonatal infant mortality (mortality in children aged 28 to 364 days) is higher than in older children, and in sub-Saharan Africa mortality in this age group is largely infectious in nature.[9] Any effect of azithromycin for prevention of child mortality is likely achieved through reductions in infectious mortality.[10] Distribution of azithromycin during early infancy when the risk of infectious mortality is highest may prove to be an efficacious strategy for targeting azithromycin treatment for child survival. Here, we describe an individually randomised placebo-controlled double-masked trial designed to establish the efficacy of administration of azithromycin during the neonatal period for the prevention of child mortality. Our central hypothesis is that a single oral dose of azithromycin administered during the neonatal period will reduce all-cause infant mortality compared with a single dose of placebo.

## METHODS/DESIGN
### Study design
The *Nouveaux-nés et Azithromycine: une Innovation dans le Traitement des Enfants* (NAITRE) study is an individually randomised placebo-controlled trial designed to evaluate the efficacy and safety of administration of a single dose of oral azithromycin during the neonatal period for the prevention of infant mortality (figure 1). Neonates aged

8–27 days are randomised in a 1:1 fashion to a single dose of oral azithromycin or matching placebo and followed for 12 months. The primary outcome of the trial is 6-month infant mortality (figure 2). Enrolment commenced in April 2019 and is expected to last until mid-2022.

### Objective and hypothesis
The objective of this study is to establish the safety and efficacy of administration of a single dose of oral azithromycin to neonates aged 8–27 days for the prevention of infant mortality. We hypothesise that neonates randomised to azithromycin will have significantly lower all-cause mortality by 6 months of age compared with those randomised to placebo.

### Study oversight
An independent Data and Safety Monitoring Committee (DSMC) oversees data collection and patient safety in this study. The DSMC contains members with expertise in paediatrics, infectious disease, biostatistics, bioethics and mass azithromycin distribution. The DSMC meets annually in a face-to-face meeting, including once prior to commencement of the study. Quarterly phone meetings during the implementation of the trial take place to monitor progress and review safety data. In addition to the DSMC, a trial steering committee headquartered in Burkina Faso annually reviews the progress of the meeting. The steering committee meets annually to review trial progress and consists of key stakeholders in

| | STUDY PERIOD | | | | | |
|---|---|---|---|---|---|---|
| | Enrolment | Allocation | Post-allocation | | | Close-out |
| **TIMEPOINT** | | *0* | *Weeks 1, 2, 3* | *3 months* | *6 months* | *12 months* |
| **ENROLMENT:** | | | | | | |
| **Eligibility screen** | X | | | | | |
| **Informed consent** | X | | | | | |
| **Allocation** | | X | | | | |
| **INTERVENTIONS:** | | | | | | |
| **Treatment (oral azithromycin or placebo)** | | X | | | | |
| **ASSESSMENTS:** | | | | | | |
| **Baseline questionnaire** | X | | | | | |
| **Anthropometry** | X | | | | X | |
| **Adverse events** | | | X | X | | |
| **Vital status** | | | X | X | X | X |

**Figure 2** SPIRIT schedule of enrolment, interventions and assessments.

policy, child health and research in Burkina Faso. Serious adverse events are reviewed by two medical monitors, one based in Burkina Faso and one in the USA.

## Setting
This study is taking place in nine regions throughout Burkina Faso. All study sites are within a 4-hour drive of a facility with capability of performing pyloromyotomy (Centre Hospitalier Universitaire Pédiatrique Charles de Gaulle in Ouagadougou or Centre Hospitalier Universitaire Souro Sanou in Bobo Dioulasso). Study sites are situated in peri-urban (eg, outside of Ouagadougou) or rural areas. Pregnancies, births and newborns are identified and enrolled at a *Centre de Santé et de Promotion Sociale* (CSPS) in the study area. CSPSs are government nurse-led primary care facilities that offer preventive and curative care to catchment areas covering several rural villages.[11] Study headquarters are at the Centre de Recherche en Santé de Nouna in Nouna, northwestern Burkina Faso. All study personnel underwent didactic and hands-on training in good clinical practice, trial conduct and trial procedures. Study supervisors and investigators regularly review study data and conduct site visits to ensure adherence to the trial protocol.

## Recruitment
Newborns are recruited through several mechanisms, including identification of pregnant women in their third trimester, facility-based births, key informant or community health worker systems in study areas, and through vaccination days. Pregnant women are informed about the study and asked to contact study staff after delivery if they are interested in their child participating in the trial.

Women who give birth in facilities participating in the study are approached prior to discharge and are asked to return with their infant in 1–2 weeks for eligibility assessment and enrolment. Verbal consent is obtained from pregnant women and women who have recently delivered to collect contact information, and to follow their pregnancy and contact them when the child is in the eligible window for enrolment if they are interested in their child participating in the study. Caregivers and newborns are also referred to the study by community health workers and key informants who work in the communities in the catchment areas for participating study clinics. Finally, women who return to health facilities with their infants for vaccination during the neonatal period are approached about the study if their child is in the eligible age range. In this setting, the BCG vaccination is typically given in the weeks following birth on a specified 'BCG vaccination day', and caregivers attending vaccination days are thus informed about the study. Caregivers of all children in the eligible age range are informed about the study.

## Inclusion and exclusion criteria
Newborns are eligible to be enrolled if they are from 8 to 27 days of age, weigh at least 2500 g at the time of enrolment, have no known allergy to azalides, no neonatal jaundice potentially indicating hepatic failure and are able to feed orally. Children who are too young or too small at initial evaluation can return for a second evaluation for inclusion in the trial as long as they are in the study's enrolment age range. While lower weight (which may be indicative of younger gestational age, low birth weight or failure to thrive) children may benefit from the azithromycin

intervention, they are excluded from the trial due to mixed evidence of the role of birth weight and gestational age on risk of IHPS, some of which indicates that smaller babies are at increased risk of IHPS.[12] In addition, the family of the newborn must intend to stay in the study area for at least 6 months to minimise loss to follow-up and appropriate consent must be obtained from the caregiver.

### Enrolment and baseline survey

After screening and prior to formal enrolment, written informed consent is obtained from the caregiver of all participants. Participants are then enrolled into the study and assigned a study identification number. The baseline assessment includes anthropometry (weight, length and mid-upper arm circumference measurement), recording of the child's birth weight (obtained from the child's health card), timing of initiation of breastfeeding and pregnancy type (singleton or multiple).

### RANDOMISATION

Children are randomised in a 1:1 fashion to a single dose of azithromycin or placebo. The randomisation list was generated by TCP and WWG in R V.3.3. Treatment letters that correspond to azithromycin or placebo were randomly assigned to study identification numbers. Children are assigned via the study's mobile application to a study identification number and thus a randomisation letter. To prevent accidental unmasking, a total of eight letters are used in the study, with four referring to azithromycin and four referring to placebo. Treatment bottles are labelled with study-specific labels that are identical in appearance with the exception of the treatment letter to facilitate masking.

### Treatment and masking

Treatment is provided as a single, oral, directly observed dose of azithromycin (approximately 20 mg/kg, weight-based dosing) or matching placebo. Azithromycin for oral suspension is provided in bottles containing azithromycin dehydrate powder equivalent to 1200 mg per bottle. The placebo oral suspension is identical to the azithromycin with the exception of the active ingredient to achieve masking. After children are weighed as part of the baseline anthropometric assessment, the tablet will automatically calculate the dosage of treatment. All study medications, including azithromycin and matching placebo, are donated by Pfizer.

### Allocation concealment

Allocation concealment is achieved via masking and the electronic tablet. With a fully masked trial, study staff, investigators and participants are not aware of which assignment they will be given or have received. Furthermore, study staff are not aware of the randomisation letters that will be assigned to the next participant until that participant has undergone screening, informed consent, baseline assessment and randomisation. After

the baseline visit is complete, the tablet displays the treatment letters corresponding to the medication bottle with which the child will be treated. Viewing treatment letters prior to completion of the baseline form is not possible on the tablet, thus achieving allocation concealment.

### Follow-up

All participants complete follow-up visits weekly for 3 weeks after treatment and then at 3, 6 and 12 months of age. The weekly adverse event follow-up visits and the 3-month and 12-month visits occur via phone call or home visit. The 6-month visit is an in-person visit conducted at the CSPS. To minimise loss to follow-up, several forms of contact information are obtained for each participant as well as their location of residence. Community health workers and key informants help facilitate follow-up visits, and home visits are conducted in case if participants are not contactable. For the primary outcome, a child's vital status must be known to be considered not to lost follow-up (eg, the child is known to be alive at 6 months of age or known to have died at or before the 6-month visit). Information regarding whether the child was alive, died, had moved or their status was unknown is recorded at each time point.

### Primary outcome measurement

The primary outcome is all-cause mortality determined at the 6-month study visit. Vital status is measured via interview with the caregiver or head of household. Children who are living are evaluated in person at the CSPS or via home visit. Vital status is recorded in the study's mobile application via tablet.

### Secondary outcome measurements

*12-month mortality.* Vital status is measured via caregiver interview determined at 12 months of age. The 12-month study visit is conducted via phone call or in-person visit.

*Neonatal mortality.* As a prespecified secondary outcome, we will assess neonatal mortality (mortality prior to 28 days of age) by study arm. Vital status will be measured in all children at each weekly follow-up visit for 3 weeks, and as such 28-day mortality will be measured in all children regardless of age of enrolment. Although we do not anticipate that this secondary analysis will be fully powered, it is prespecified to assess if azithromycin dosing during the neonatal period has a large effect on neonatal mortality.

*Hospitalisation.* At each study visit, caregivers are asked if they had sought medical care for their child since they last spoke with the study team, and if so if their child had stayed overnight in a healthcare facility.

*Anthropometric outcomes.* At the 6-month study visit, anthropometric measurements including weight, length and mid-upper arm circumference are collected from all children. Weight gain from baseline in g/kg/day will be calculated to estimate weight gain velocity by study arm. Weight gain velocity in g/kg/day is a commonly used metric for weight gain, nutritional status over time and identification of growth deficits in newborns.[13 14] Length

gain in mm/day will be calculated by study arm. Weight-for-height Z-score (to measure wasting), height-for-age Z-score (to measure stunting) and weight-for-age Z-score (to measure underweight) will be calculated according to 2006 WHO standards.[15]

*Cause-specific mortality.* A verbal autopsy interview will be undertaken with the caregiver of all children who die during the course of the study using the 2014 WHO verbal autopsy instrument or the neonatal verbal autopsy instrument if the death occurs before 4 weeks of age. Cause of death data will be analysed using the InterVA algorithm.[16 17]

### Adverse events

The primary adverse event under surveillance is IHPS. Active surveillance for IHPS and other adverse events occurs via weekly screening of all enrolled children for 3 weeks after treatment and at 3 months of age. Although evidence of timing of potentially macrolide-related IHPS is limited, previous studies have documented that pyloromyotomy was performed 2–4 weeks following exposure,[4] with symptoms developing prior to surgical correction. IHPS development is rare in children older than 6 weeks of age and is not expected to occur after 3 months of age.[4 18] In addition, during the informed consent process the caregiver of all enrolled children is informed of the signs and symptoms of IHPS and asked to contact study staff should the child exhibit abnormal vomiting. All caregivers are given a fact sheet with information about IHPS and its signs and symptoms. Children are enrolled in facilities where they would seek care if necessary, and all healthcare personnel in the facilities are aware of study activities. If a child enrolled in the study is brought to the facility, study staff are informed. If active or passive surveillance identifies a child with projectile or progressive vomiting that occurs consistently after eating (ie, the child is unable to eat anything without vomiting), the child will be referred to the district hospital for evaluation of clinical symptoms and possible referral to the regional or national hospital for an ultrasound. Any child with projectile vomiting is followed until resolution. If the ultrasound indicates that the pylorus is hypertrophied per Burkinabé national guidelines (>4 mm for pyloric muscle thickness or >15 mm for pyloric length) or the measurements are within normal limits but no food can pass per dynamic ultrasound, the child will be transferred to a national hospital for further evaluation, diagnosis and pyloromyotomy if IHPS is diagnosed. Children diagnosed with IHPS will be followed at 1 week and 4 weeks after surgery to ascertain surgical outcomes and then followed according to the study schedule.

In addition to screening for vomiting and IHPS, caregivers are interviewed weekly for 3 weeks following treatment for adverse events including rash, diarrhoea and fever. Caregivers are asked if they sought medical care for their child or if the child was hospitalised since the last visit by the study team and if so the indication for seeking treatment or hospitalisation. Finally, verbal autopsy is performed for all deaths to ascertain cause of death. The study's DSMC reviewed and approved the adverse event monitoring plan prior to the start of recruitment and reviews all reports of IHPS in real time. Our adverse event monitoring protocol could be modified per DSMC recommendation if, for example, IHPS was more common than anticipated. Although the DSMC reviews each IHPS case in real time in a masked fashion and study data in aggregate quarterly, they may request IHPS or other safety data by arm if there are concerns related to safety. Each suspected case of IHPS is reviewed by two medical monitors and all suspected cases are followed until resolution.

### Interim analysis

One prespecified interim analysis for the primary outcome is planned when full data are available for one-third of the enrolments or after one full year of enrolment (eg, 6 months after the last individual), whichever comes first. A p value of <0.001 is the prespecified stopping rule for efficacy for this interim analysis. Thus the final analysis will be conducted at an alpha of 0.049.

### Statistical analysis

The primary analysis will be conducted as a binomial regression model with a complementary log-log link, with inferences based on a Monte Carlo permutation test based on the randomisation unit. The complementary log-log link allows for estimation of the relative hazard and can account for differing follow-up time. As a prespecified subgroup analysis, we will assess effects of age at time of treatment in days using a binomial regression model, complementary log-log link and an interaction for treatment arm by age. All statistical tests for efficacy will be two-sided. A p value of <0.05 will be considered statistically significant.

### Sample size considerations

Assuming a mortality probability of 35 per 1000 and loss to follow-up of 10%, a sample size of 10 856 per arm (21 712 total) would yield approximately 80% power to detect a 20% decrease in mortality in children randomised to azithromycin compared with placebo.

### Data management

Data are collected via smartphone in the field using Survey Solutions (World Bank Group, Washington, DC, USA), a cloud-based platform for electronic data collection. Smartphones are synced in the field to a cloud-based server.

### Patient and public involvement

Members of the community, including key informants in the study communities, are involved in identification of births, recruitment of children and facilitating follow-up visits. Results will be disseminated in collaboration with community leadership in study areas. Patients and the public were not involved in the design or planning of the study.

### Ethics and dissemination

Written informed consent is obtained from the caregiver of each enrolled child (online supplementary file l). Efficacy and safety results of this study will be disseminated to community, policy and scientific stakeholders, including the Ministry of Health in Burkina Faso, the WHO and relevant non-governmental organisations that implement child health policies. Results will be shared with the scientific community via publication in peer-reviewed journals and presentation at international conferences.

## DISCUSSION

Although biannual mass azithromycin distribution has been shown to reduce all-cause child mortality in children aged 1–59 months,[12] this strategy may not reach the youngest children who may benefit the most. With biannual community distributions, children would first be reached any time between the age of 1 and 7 months or on average at approximately 4 months of age. Several strategies exist for reliably reaching younger children. For example, quarterly mass treatment would reach children at a younger age on average. Targeted treatment of young children is a strategy that may integrate well with existing health systems and can allow for reliably treated children at specific ages. In Burkina Faso, healthcare for children under five is free of charge, and as a result healthcare coverage for children is high. For example, vaccination coverage reportedly exceeded 80% in 2014 in Nouna.[19] If shown to be both efficacious and safe, targeting treatment to the youngest, highest-risk children could be considered via integration with existing well-child visits in the health system.

IHPS is a rare but serious condition with approximately 2 cases per 1000 infants in the USA.[4] Surgical treatment is required. However, in many regions of sub-Saharan Africa, infant mortality rates far exceed the risk of IHPS.[20] Prevention of a portion of infant mortality in the highest risk areas may therefore offer substantial benefits for populations. However, the associated risks of IHPS as well as the efficacy must be well understood before any such policy could be put into place. In this study, a comprehensive screening and referral protocol are in place to identify any cases of IHPS that arise during the course of the study. We expect that the results of this study will provide evidence of the risks and benefits of treatment of neonates with azithromycin for prevention of child mortality.

To date, existing evidence for the relationship between macrolide use and development of IHPS has been limited to observational studies or small randomised trials of azithromycin for prevention of bronchopulmonary dysplasia in low birth weight infants.[21–23] In a large retrospective cohort study in the USA, among 4875 infants up to 90 days of life, 8 cases of IHPS occurred, with a higher risk of IHPS among children receiving erythromycin or azithromycin compared with cephalexin.[4] This risk appeared to be higher in children receiving erythromycin compared

with azithromycin, and among children under 14 days of age compared with older children. Observational studies are limited by confounding by indication, which could occur if children with conditions necessitating a macrolide have different conditions or are otherwise substantively different from those who receive another antibiotic class or no antibiotics. Existing randomised controlled trials have been small (total n=263 infants randomised), and no cases of IHPS have been detected. Relatively little evidence of the epidemiology of IHPS in sub-Saharan Africa exists,[24–26] although some have hypothesised that IHPS is less common in developing country settings compared with high-income settings due to different feeding and medication practices.[26–28] The results of this study are expected to provide comprehensive, rigorous evidence of the relationship between azithromycin use and IHPS in neonates, as well as important data on the epidemiology of IHPS in rural West Africa.

Several limitations must be considered in this study. This study was designed under the principle of a large simple trial, due to the large sample size required to be adequately powered for a rare outcome.[29 30] However, careful attention must be paid to each individual in the trial due to the risk of IHPS. The focus is on measuring the most important outcomes (mortality and IHPS), and thus measurement of additional secondary outcomes is limited. For example, the potential for antimicrobial resistance is an important consideration with the use of azithromycin for child health.[31 32] Parallel individual and cluster randomised trials are currently evaluating the effect of azithromycin on selection for macrolide resistance in neonates and older children (ClinicalTrials.gov NCT03676751 and NCT03676764). Given the absence of IHPS data from rural sub-Saharan Africa and the expected rarity of the event, the study may be underpowered to detect differences in IHPS between study arms. Stopping rules for safety are based not only on statistical significance, but decisions will be made in conjunction with the DSMC based on case reports arising from the study. The results of this study are expected to be generalisable to neonates in similar West African settings, but results may not be generalisable outside of rural West Africa, where infectious aetiology leading to infant mortality may be different. Results may also not be generalisable to low-weight neonates, as they were excluded from the trial. We anticipate that the study will provide the largest sample to date providing evidence of the epidemiology of IHPS in West Africa, as well as the largest trial to date evaluating azithromycin for IHPS.

## CONCLUSIONS

The results of this study are expected to inform policy related to the use of azithromycin for prevention of child mortality by specifically evaluating the role of targeting doses during early infancy when the risk of infectious mortality is highest. This work is anticipated to build on the evidence base arising from the MORDOR study, which

demonstrated that biannual mass treatment significantly reduces all-cause child mortality in children aged 1–59 months. If proven to be safe and effective, the evidence arising from this study could be used to develop policies related to administration of azithromycin during the neonatal period.

**Collaborators** Clinical Centers, Committees, and Resource Centers for the NAITRE Study. Clinical Centers: Centre de Recherche en Sante de Nouna, Nouna, Burkina Faso: Ali Sie (Principal Investigator), Mamadou Bountogo, Mamadou Ouattara, Boubacar Coulibaly, Cheik Bagagnan, Alphonse Zakane, Guillaume Compaoré, Valentin Boudo, Lucienne Ouermi, Aristide Ouedraogo, Ouedraogo Andiyam Thierry, Eric Nebie; Centre Hospitalier Universitaire Pédiatrique Charles De Gaulle, Ouagadougou, Burkina Faso: A.M. Napon; Centre Hospitalier Souro Sanou, Bobo Dioulasso, Burkina Faso: Z. Nikièma. Committees: Data and Safety Monitoring Committee: Allen Hightower (Chair), Amza Abdou, Miriam Laufer, Jacqueline Glover, and Wafaie Fawzi. Resource Centers: Trial Coordinating Center, F.I. Proctor Foundation, University of California, San Francisco, California, USA: Jessica Brogdon, Catherine Cook, Thuy Doan, William W. Godwin, Jeremy D. Keenan, Elodie Lebas, Thomas M. Lietman (Principal Investigator), Ying Lin, Kieran S. O'Brien, Catherine E. Oldenburg (Principal Investigator), Travis C. Porco. Heidelberg University, Heidelberg Germany: Till Bärnighausen.

**Contributors** Conceived and designed study: AS, MB, EN, MO, BC, CB, PZ, EL, JB, WWG, YL, TP, TD, TML, CEO. Created tables/figures: AS, EL, JB, CEO. Wrote first draft of manuscript: CEO. Read and critically revised manuscript: AS, MB, EN, MO, BC, CB, PZ, EL, JB, WWG, YL, TP, TD, TML, CEO. Members of the NAITRE Study Group are listed in the supplementary file.

**Funding** The NAITRE Study is funded by the Bill and Melinda Gates Foundation (OPP1187628). CEO and TD were supported in part by a Research to Prevention Blindness Career Development Award. The funders played no role in the design, writing or decision to publish study protocol.

**Competing interests** Study medication (azithromycin and placebo) are donated by Pfizer (New York, NY).

**Patient consent for publication** Not required.

**Ethics approval** Ethical approval was obtained for the NAITRE Study from the Institutional Review Boards at the University of California, San Francisco (Protocol #18-25027) and the Comité National d'Ethique pour la Recherche (National Ethics Committee of Burkina Faso; Protocol #2018-10-123).

**Provenance and peer review** Not commissioned; externally peer reviewed.

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
