## [Reviewer comments · BMJ Open]

ARTICLE DETAILS

TITLE (PROVISIONAL)	Neonatal azithromycin administration to prevent infant mortality: study protocol for a randomized controlled trial
AUTHORS	Sie, Ali; Bountogo, Mamadou; Nebie, Eric; Ouattara, Mamadou; Coulibaly, Boubacar; Bagagnan, Cheik; Zabre, Pascal; Lebas, Elodie; Brogdon, Jessica; Godwin, William; Lin, Ying; Porco, Travis; Doan, Thuy; Lietman, Thomas; Oldenburg, Catherine

VERSION 1 – REVIEW

REVIEWER	Abhik Das RTI International
REVIEW RETURNED	31-May-2019

GENERAL COMMENTS	This is a well written and competently described research protocol for a randomized trial that needs to be done. A few comments on the protocol are below: 1. Is 3 weeks follow up for safety adequate? This may need to be better justified, using the expected timing for any IHPS occurrence.2. The exclusion criteria, while intended to achieve a clear signal, also result in the sickest babies that may benefit most from such an intervention, being excluded. If successful, would the results of this trial be applied to such babies?3. Anthropometry outcomes: growth velocity can also be measured using WHO standards.4. Since IHPS is a significant safety concern, it seems that more intensive and frequent interim safety monitoring is warranted, so that the DSMC can recommend stopping the trial for safety if necessary.5. Statistical analysis: why not use robust Poisson regression instead of logistic regression, because Relative Risks are more interpretable and much better understood and intuitive than Odds Ratios.
--

REVIEWER	Robert L. Schelonka Oregon Health and Sciences University
REVIEW RETURNED	04-Jun-2019

GENERAL COMMENTS	The manuscript titled: Neonatal azithromycin administration to prevent infant mortality: design and rationale of a randomized controlled trial by Ali Sié and colleagues (including contact author
--

	Dr. Oldenburg) is a description of a planned RCT taking place in sub-Saharan Africa. The outcome, mortality at 6 months, is measurable, appropriate and important. The sample size includes >20,000 newborn infants. Administering the medication or placebo to study subjects will be practically challenging. Even more so will be tracking and following the follow up data at healthcare visits. The authors articulate a logical plan to do so as well as examining vital statistics to capture all deaths for the first 6 months of age. Also important will be to evaluate patients for medication side effects, most notably development of hypertrophic pyloric stenosis. It would be interesting to determine if there were changes in antimicrobial resistance patterns after mass treatment of newborn infants with azithromycin. After reading the manuscript, I am intrigued by this study. If effective regarding lives saved, it would be most illuminating to learn about downstream effects of such widespread antibiotic administration on side effects including pyloric stenosis or antimicrobial resistance.
--	---

REVIEWER	Jacqz-Aigrain Evelyne Assistance publique Hopitaux de Paris
REVIEW RETURNED	25-Jun-2019

GENERAL COMMENTS	The study is ambitious, will last 2 or more years with individual child and family follow-up of 1 year: organization and visits are well planned but follow-up will be complex. Training of all health care professionals for trial conduct and GCP will be key issues for success. . Exclusion criteria are limited : what about neonates with diseases (infection..., other complications, malformations ...) in the first week of life ? Is the study aiming to propose the study to all families and include "all neonates" in the different recruiting centers whatever the perinatal events ? Baseline data might be more detailed. . The risk of IHPS is low, but what about the other side-effects : cutaneous, convulsions, QT prolongation ...will a list of potential side effects be establish to facilitate collection of information ? . Neonatal jaundice is not hepatic failure ? . The planned drop out is only 10% : how was this defined as it seems low for this study. What are the plans for missing visits : during the trial, what will be done to find a family that did not come to a follow-up visit ? at the end of the trial, how many missing visits and calls will be "acceptable" for a patient, for the whole study ?
---

VERSION 1 – AUTHOR RESPONSE

Reviewer: 1

Reviewer Name: Abhik Das

Institution and Country: RTI International

Please state any competing interests or state 'None declared':None declared

Please leave your comments for the authors below

This is a well written and competently described research protocol for a randomized trial that needs to be done. A few comments on the protocol are below:

RESPONSE: We thank the reviewer for their comments.

1. Is 3 weeks follow up for safety adequate? This may need to be better justified, using the expected timing for any IHPS occurrence.

RESPONSE: Limited observational data exists of IHPS following macrolide exposure, in which surgery to correct IHPS was performed approximately 2-4 weeks following macrolide exposure (vomiting symptoms would occur earlier than this). IHPS is uncommon after 6 weeks of age (typically symptoms develop in the first few weeks of life) and does not occur in infants over 3 months of age. We have an additional 3-month safety visit, and no IHPS is expected to occur after 3 months of age. We have added additional justification for the 3-weekly and 3-month visits (Page 11, Line 181):

“Active surveillance for IHPS and other adverse events occurs via weekly screening of all enrolled children for three weeks after treatment and at 3 months of age. Although evidence of timing of potentially macrolide-related IHPS is limited, previous studies have documented that pyloromyotomy was performed 2-4 weeks following exposure⁴, with symptoms developing prior to surgical correction. IHPS development is rare in children older than 6 weeks of age and is not expected to occur after 3 months of age.^{4,15}”

2. The exclusion criteria, while intended to achieve a clear signal, also result in the sickest babies that may benefit most from such an intervention, being excluded. If successful, would the results of this trial be applied to such babies?

RESPONSE: We exclude babies who are <2500 g at the time of treatment out of an abundance of caution for IHPS (e.g., to screen out babies who were very low birthweight, who might be at increased risk of IHPS) and those who cannot feed orally because they would be unable to take the study medication. We agree that the lower weight children may benefit from an intervention such as azithromycin, however as this will be one of the first studies of azithromycin administration in the neonatal period, we chose to err on the side of caution with smaller babies. We note that if the baby is too small at day 8, they can come back up to day 27 when they are slightly larger and be eligible to be included in the study. We have added additional details to the Methods (Page 7, Line 90):

“Children who are too young or too small at initial evaluation can return for a second evaluation for inclusion in the trial as long as they are in the study’s enrollment age range. While lower weight (which may be indicative of younger gestational age, low birthweight, or failure to thrive) children may benefit from the azithromycin intervention, they are excluded from the trial due to mixed evidence of the role of birthweight and gestational age on risk of IHPS, some of which indicates that smaller babies are at increased risk of IHPS.¹²”

We have also added a discussion of generalizability to the Discussion (Page 17, Line 312):

“The results of this study are expected to be generalizable to neonates in similar West African settings, but results may not be generalizable outside of rural West Africa, where infectious etiology leading to infant mortality may be different. Results may also not be generalizable to low-weight neonates, as they were excluded from the trial.”

3. Anthropometry outcomes: growth velocity can also be measured using WHO standards.

RESPONSE: We agree, although we note they measure slightly different things. We will analyze the WHO standards as well as weight gain velocity in g/kg/day, which is a commonly used metric for measuring weight gain in infants. We have added references for the use of g/kg/day and also included the WHO standards in our anthropometric outcomes section (Page 11, Line 169):

“Weight gain velocity in g/kg/day is a commonly used metric for weight gain, nutritional status over time, and identification of growth deficits in newborns.^{13,14} Length gain in mm/day will be calculated by study arm. Weight-for-height Z-score (to measure wasting), height-for-age Z-score (to measure stunting), and weight-for-age Z-score (to measure underweight) will be calculated according to 2006 World Health Organization (WHO) standards.¹⁵”

4. Since IHPS is a significant safety concern, it seems that more intensive and frequent interim safety monitoring is warranted, so that the DSMC can recommend stopping the trial for safety if necessary.

RESPONSE: We agree with this comment and realize we were not clear with our full interim safety monitoring plan. While we only have one formal pre-specified interim analysis, the DSMC is carefully reviewing every case of IHPS that is reported in the study. If an excess number of IHPS events were to be reported, the DSMC will ask for IHPS cases by study arm. Our DSMC is reviewing the records of any children who are diagnosed with IHPS in real time (to date, there have been no IHPS cases). In addition, two medical monitors review each report of suspected IHPS and each suspected case is followed carefully until resolution. The DSMC additionally reviews data in aggregate quarterly. We have clarified this in the Methods (Page 13, Line 209):

“The study’s DSMC reviewed and approved the adverse event monitoring plan prior to the start of recruitment and reviews all reports of IHPS in real time. Our adverse event monitoring protocol could be modified per DSMC recommendation if, for example, IHPS was more common than anticipated. Although the DSMC reviews each IHPS case in real-time in a masked fashion and study data in aggregate quarterly, they may request IHPS or other safety data by arm if there are concerns related to safety. Each suspected case of IHPS is reviewed by two medical monitors and all suspected cases are followed until resolution.”

5. Statistical analysis: why not use robust Poisson regression instead of logistic regression, because Relative Risks are more interpretable and much better understood and intuitive than Odds Ratios.

RESPONSE: We have changed the primary analysis to binomial regression with a complementary log-log link. This model allows for estimation of the relative hazard and can adjust for different follow-up times. Given the rarity of the primary outcome (mortality), we anticipate that the odds ratio and the relative risk will be almost identical, although we agree that relative risks are more interpretable than odds ratios. We have edited the Methods (Page 13, Line 224):

“The primary analysis will be conducted as a binomial regression model with a complementary log-log link, with inferences based on a Monte Carlo permutation test based on the randomization unit. The complementary log-log link allows for estimation of the relative hazard and can account for differing follow-up time. As a pre-specified subgroup analysis, we will assess effects of age at time of treatment in days using a binomial regression model, complementary log-log link, and an interaction for treatment arm by age. All statistical tests for efficacy will be two-sided. A P-value of <0.05 will be considered statistically significant.”

Reviewer: 2

Reviewer Name: Robert L. Schelonka

Institution and Country: Oregon Health and Sciences University

Please state any competing interests or state ‘None declared’: None

Please leave your comments for the authors below

The manuscript titled: Neonatal azithromycin administration to prevent infant mortality: design and rationale of a randomized controlled trial by Ali Sié and colleagues (including contact author Dr.

Oldenburg) is a description of a planned RCT taking place in sub-Saharan Africa. The outcome, mortality at 6 months, is measurable, appropriate and important. The sample size includes >20,000 newborn infants. Administering the medication or placebo to study subjects will be practically challenging. Even more so will be tracking and following the follow up data at healthcare visits. The authors articulate a logical plan to do so as well as examining vital statistics to capture all deaths for the first 6 months of age. Also important will be to evaluate patients for medication side effects, most notably development of hypertrophic pyloric stenosis. It would be interesting to determine if there were changes in antimicrobial resistance patterns after mass treatment of newborn infants with azithromycin.

RESPONSE: We thank the reviewer for their comments. We agree that measuring antimicrobial resistance is an important outcome in studies of azithromycin for child survival. This trial is one in a program of four trials currently ongoing or planned in Burkina Faso evaluating the use of azithromycin for neonatal, infant, and child survival. Although we were unable to include AMR measurements in this large simple trial (as it was practically impossible due to the sample size and logistical challenges for achieving our primary outcome that the reviewer notes), we are currently in the process of starting another trial which will include detailed swab collections in neonates, infants, and children randomized to a single dose of azithromycin or placebo and followed longitudinally (clinicaltrials.gov NCT03676751). We have added to the Discussion (Page 17, Line 304):

“For example, the potential for antimicrobial resistance is an important consideration with the use of azithromycin for child health.^{31,32} Parallel individual and cluster randomized trials are currently evaluating the effect of azithromycin on selection for macrolide resistance in neonates and older children (clinicaltrials.gov NCT03676751 and NCT03676764).”

After reading the manuscript, I am intrigued by this study. If effective regarding lives saved, it would be most illuminating to learn about downstream effects of such widespread antibiotic administration on side effects including pyloric stenosis or antimicrobial resistance.

RESPONSE: We thank the reviewer for their comments.

Reviewer: 3

Reviewer Name: Jacqz-Aigrain Evelyne

Institution and Country: Assistance publique Hopitaux de Paris

Please state any competing interests or state 'None declared': No competing interests

Please leave your comments for the authors below

The study is ambitious, will last 2 or more years with individual child and family follow-up of 1 year: organization and visits are well planned but follow-up will be complex. Training of all health care professionals for trial conduct and GCP will be key issues for success.

RESPONSE: We thank the reviewer for their comments, and we agree about the complexity of follow-up visits. We have included more details below about our efforts to maintain high follow-up rates (which are currently well above targets). All health care professionals involved in the study undergo extensive didactic and hands-on training in preparation for the trial and also have ongoing supervision by study staff and investigators. We have added to the Methods (Page 6, Line 65):

“All study personnel underwent didactic and hands-on training in good clinical practice, trial conduct, and trial procedures. Study supervisors and investigators regularly review study data and conduct site visits to ensure adherence to the trial protocol.”

. Exclusion criteria are limited : what about neonates with diseases (infection..., other complications, malformations ...) in the first week of life ? Is the study aiming to propose the study to all families and include “all neonates” in the different recruiting centers whatever the perinatal events ? Baseline data might be more detailed.

RESPONSE: We are enrolling children in the 2nd through 4th week of life. Children that have adverse birth outcomes that result in mortality during the first week will not be eligible by definition. We are recruiting all-comers based on birth registries from the health facilities and vaccination visits (for example, BCG vaccination in this setting is typically given on a “BCG vaccination day” where newborns come back to the health facility during the first few weeks of life, rather than at birth, and we recruit a large percentage of participants from this vaccination visit). We have added additional details about recruitment (Page 7, Line 81):

“Finally, women who return to health facilities with their infants for vaccination during the neonatal period are approached about the study if their child is in the eligible age range. In this setting, the bacille Calmette-Guérin (BCG) vaccination is typically given in the weeks following birth on a specified “BGC vaccination day”, and caregivers attending vaccination days are thus informed about the study. Caregivers of all children in the eligible age range are informed about the study.”

. The risk of IHPS is low, but what about the other side-effects : cutaneous, convulsions, QT prolongation ...will a list of potential side effects be establish to facilitate collection of information ?

RESPONSE: Caregivers are interviewed about a number of potential side effects, including rash, gastrointestinal side effects, and if they sought medical care for their child and/or the child was hospitalized, and if so, why. In this resource-limited setting, measurement of QT prolongation is not feasible. We have expanded on our adverse event monitoring section (Page 12, Line 205):

“In addition to screening for vomiting and IHPS, caregivers are interviewed weekly for three weeks following treatment for adverse events including rash, diarrhea, and fever. Caregivers are asked if they sought medical care for their child or if the child was hospitalized since the last visit by the study team and if so the indication for seeking treatment or hospitalization. Finally, verbal autopsy is performed for all deaths to ascertain cause of death.”

. Neonatal jaundice is not hepatic failure ?

RESPONSE: We agree. In this study’s setting, the only way to screen for possible hepatic failure is via neonatal jaundice. We therefore exclude babies with neonatal jaundice, even though for many of these babies it will not mean hepatic failure. We have clarified in the Methods (Page 7, Line 90):

“...no neonatal jaundice potentially indicating hepatic failure...”

. The planned drop out is only 10% : how was this defined as it seems low for this study. What are the plans for missing visits : during the trial, what will be done to find a family that did not come to a follow-up visit ? at the end of the trial, how many missing visits and calls will be “acceptable” for a patient, for the whole study ?

RESPONSE: The loss to follow-up rate is for the 6-month visit, meaning we anticipate knowing the vital status at 6 months for 90% of enrolled children (it is not a function of the number of missing visits). While we do not yet have data on follow-up at 6 months as no enrolled children have reached that time point, current 3-month follow-up data far exceeds this target. We have added additional details to the Methods on steps we are taking to minimize loss to follow-up (Page 10, Line 138):

“To minimize loss to follow-up, several forms of contact information are obtained for each participant as well as their location of residence. Community health workers and key informants help facilitate follow-up visits, and home visits are conducted in case participants are not contactable. For the primary outcome, a child’s vital status must be known to be considered not lost to follow-up (e.g., the child is known to be alive at 6 months of age or known to have died at or before the 6-month visit). Information is recorded for whether the child was alive, died, had moved, or their status was unknown at each time point.”

VERSION 2 – REVIEW

REVIEWER	Abhik Das RTI International, USA
REVIEW RETURNED	09-Aug-2019

GENERAL COMMENTS	The authors have adequately addressed all the previous comments and questions raised by this reviewer.
--

REVIEWER	Robert L. Schelonka Oregon Health and Sciences University, Portland Oregon USA
REVIEW RETURNED	07-Aug-2019

GENERAL COMMENTS	The current manuscript describes a proposed randomized controlled study examining if a single dose of azithromycin given in the early neonatal period reduces infant mortality. This is a novel approach to vitally important problem. The authors properly address the risk of medication induced hypertrophic pyloric stenosis and to a lesser degree, the development of antimicrobial resistance with widespread antibiotic use.
--